# Absolute Quantification of Pan-Cancer Plasma Proteomes Reveals Unique Signature in Multiple Myeloma

**DOI:** 10.3390/cancers15194764

**Published:** 2023-09-28

**Authors:** David Kotol, Jakob Woessmann, Andreas Hober, María Bueno Álvez, Khue Hua Tran Minh, Fredrik Pontén, Linn Fagerberg, Mathias Uhlén, Fredrik Edfors

**Affiliations:** 1Science For Life Laboratory, KTH Royal Institute of Technology, 114 28 Stockholm, Sweden; david.kotol@scilifelab.se (D.K.); jakob.woessmann@scilifelab.se (J.W.); andreas.hober@scilifelab.se (A.H.); maria.bueno@scilifelab.se (M.B.Á.); khue.hua@scilifelab.se (K.H.T.M.); linn.fagerberg@scilifelab.se (L.F.); mathias.uhlen@scilifelab.se (M.U.); 2Department of Protein Science, Division of Systems Biology, School of Chemistry, Biotechnology and Health, KTH Royal Institute of Technology, 114 28 Stockholm, Sweden; 3Rudbeck Laboratory, Uppsala University, 752 36 Uppsala, Sweden; fredrik.ponten@igp.uu.se

**Keywords:** DIA, multiple myeloma, precision medicine, targeted proteomics

## Abstract

**Simple Summary:**

A precise mass spectrometry-based method was utilized to study proteins in the blood samples of over a thousand cancer patients. By accurately identifying and measuring protein levels using mass spectrometry, we focused on multiple myeloma and found potential markers for diagnosing the disease. These markers, including the complement C1 complex, JCHAIN, and CD5L, were combined in a prediction model with high accuracy for identifying multiple myeloma patients. Our findings could significantly impact cancer research by improving diagnostic tools.

**Abstract:**

Mass spectrometry based on data-independent acquisition (DIA) has developed into a powerful quantitative tool with a variety of implications, including precision medicine. Combined with stable isotope recombinant protein standards, this strategy provides confident protein identification and precise quantification on an absolute scale. Here, we describe a comprehensive targeted proteomics approach to profile a pan-cancer cohort consisting of 1800 blood plasma samples representing 15 different cancer types. We successfully performed an absolute quantification of 253 proteins in multiplex. The assay had low intra-assay variability with a coefficient of variation below 20% (CV = 17.2%) for a total of 1013 peptides quantified across almost two thousand injections. This study identified a potential biomarker panel of seven protein targets for the diagnosis of multiple myeloma patients using differential expression analysis and machine learning. The combination of markers, including the complement C1 complex, JCHAIN, and CD5L, resulted in a prediction model with an AUC of 0.96 for the identification of multiple myeloma patients across various cancer patients. All these proteins are known to interact with immunoglobulins.

## 1. Introduction

Cancer accounts for more than ten million deaths worldwide and is considered the second most common cause of mortality today. It consists of more than 200 different subgroups, making it a very heterogeneous disease. However, disease progression typically follows the same trajectory in diverse cancers. These general features of the disease can be described as events in which cells undergo dysregulated and autonomous growth, eventually disrespecting tissue boundaries, leading to metastasis [1]. Molecular events resulting from alterations in the genome of cancer cells are critical components for identifying cancer type-specific biomarkers and have been studied and mapped for several decades [2,3,4]. Recent advances in high-throughput sequencing technologies have enabled large-scale efforts to accurately map genomic alterations across many different types and subtypes of cancer [5,6,7]. This has paved the way to study altered gene expression and immunogenic reactions, which provides new opportunities for developing biomarker panels to cover the need for early and more precise cancer diagnostics [8].

Blood plasma and the analysis of its constituents is the most general diagnostic procedure in modern medicine. It is easily accessible and provides essential information about the healthy physiological state and disease, such as cancer-induced alterations in the human body [9,10]. Alterations in gene expression offer opportunities for the early detection of circulating cancer biomarkers, which can increase the chances of long-term survival and provide a more precise diagnosis. The levels of circulating cancer biomarkers such as the prostate-specific antigen (PSA) are well studied as indicators of tissue-specific growth or damage but lack clinical specificity for cancer diagnosis [11]. This presents opportunities for novel multiplex technologies providing a way to modernize tomorrow’s clinical laboratory tests by focusing on biomarker panels with high cancer specificity. Recently, transcriptome analysis performed on cell-free RNA collected from patients with various cancers has revealed that panels of biomarkers can be used to effectively subclassify cancer types [12]. In contrast to sequencing approaches, mass spectrometry-based proteomics performed on liquid biopsies is one of the most powerful strategies for quantifying proteins in multiplex. It has rapidly developed alongside widely used sequencing technologies [13]. Liquid chromatography coupled with mass spectrometry (LC–MS/MS)-based targeted proteomics has emerged as an attractive alternative to antibody-based immunoassays due to its accurate and precise measurements of protein concentrations in complex sample matrices [14,15]. Its quantitative performance can be further enhanced by adding stable isotope standard protein epitope signature tags (SIS-PrESTs) [16] which enable absolute protein measurements needed for clinical diagnostics [17].

In this study, we performed a targeted proteomics analysis of 1800 human plasma samples utilizing 276 SIS-PrESTs towards 253 proteins providing a comprehensive plasma proteome map of 15 different cancer types. We present a view of the molecular phenotypes that distinguish patients with different cancers based on their signature plasma protein levels. The strategy for precise protein quantification was deployed utilizing SIS-PrESTs in combination with data-independent acquisition (DIA). Medium- to high-abundant blood plasma proteins were absolutely quantified, including 40 Food and Drug Administration (FDA)-approved markers. Using differential expression analysis and machine learning, the complement C1 complex, JCHAIN, and CD5L were identified as potential biomarkers for diagnosing multiple myeloma (MM).

## 2. Results

### 2.1. Cohort and Analytical Strategy

Blood plasma samples were collected from 1800 patients diagnosed with one out of eight cancers (Figure 1A). All samples were provided by the Uppsala–Umeå Comprehensive Cancer Consortium (UCAN) biobank. The broader cancer classification can be further stratified into fifteen cancer types, ranging from the most prevalent lung cancer (*n* = 289) followed by colorectal cancer (*n* = 248) down to the least prevalent pituitary neuroendocrine tumor and chronic lymphocytic leukemia (*n* = 50) (Figure 1B). All liquid biopsies were subjected to the same targeted proteomics bottom-up analytical workflow previously described [18] and were spiked with a mixture of 276 SIS-PrESTs (Appendix A) representing the same number of proteins (Figure 1C).

### 2.2. Investigated Targets and Analytical Performance

An assay covering 1013 peptides from 253 proteins was established using spiked SIS-PrESTs in a plasma background. In total, 146 proteins were absolutely quantified spanning more than six orders of magnitude in concentration range (Appendix A) with 395 SIS-PrEST peptides. Quantified proteins included 40 FDA- and 16 Clinical Laboratory Improvement Amendments (CLIA)-approved clinical targets. Moreover, 23 protein members of the complement cascade were part of the analysis (Figure 2A). Within the quantified proteins, 128 (51%) were actively secreted proteins in blood according to the annotation of the human secretome [19]. To assess the performance and reproducibility of the targeted assay, we determined the intra-assay variation using pool samples randomly distributed onto all plates. The median intra-assay coefficient of variation (CV) was between 7.16 and 20% CVs (median CV = 11.3%) per plate and 17.2% CV across all 31 plates. The pool samples across all plates were highly correlated with a median Pearson’s r of 0.99 (Figure 2B). The overall biological variation between all subjects was low, with a median normalized IQR = 1.7 (Figure 2C). This signifies that even with the large variety of cancer types, most of the absolute protein variation was less than two-fold when compared between different cancers. However, two proteins with interindividual variation were observed. Those were pregnancy zone protein (PZP) and apolipoprotein(a) (LPA). The large difference in PZP was observed due to its 10-fold higher concentration in the female population compared to males. The LPA variability was caused by its quantification peptide overlapping with a repeated kringle domain whose count is dependent on the individual’s genotype, as reported in [20]. Overall, it was possible to report absolute concentration measurements of plasma proteins with low bias with respect to their levels.

### 2.3. Identification of Potential Biomarkers

Differential expression analysis was performed comparing each cancer type against the rest on the peptide level (Figure 3A). In cases of male- and female-specific cancers, only the patients of the relevant sexes were compared. We could observe that in cases where proteins were quantified with multiple peptides, all of them were significantly downregulated or upregulated. Here, cases such as the platelet basic protein (PPBP) in acute myeloid leukemia, which was previously identified as a dysregulated hub gene, can be highlighted [21]. Other examples are C1qB and C1qC, for which four peptides were identified as significantly downregulated in MM. All the targets identified as differentially expressed have been summarized in a protein network (Figure 3B). In further investigation, we focused on the unique protein pattern of MM patients, which formed an isolated island of four proteins, all part of the complement C1 complex. 

### 2.4. Downregulation of Components of the C1 Complex in Multiple Myeloma

Multiple myeloma, characterized by its heterogeneous nature as a hematologic malignancy, displays dysregulation within the plasma proteome due to the accumulation of plasma cells within the bone marrow, ultimately displacing healthy blood cells. In the context of this study, a profound alteration in the plasma proteome of multiple myeloma patients was observed in the label free MS data acquired alongside the absolute quantification MS data. Notably, this alteration stemmed from a widespread reduction in peptides linked to IGHM and IGHA1 as seen in the volcano plot (Appendix A). The label free data also showed a couple of non-IgG related proteins that undergo dysregulation in multiple myeloma, among them being albumin and APOA1, aligning with established literature [22]. 

The diagnosis of multiple myeloma relies on the detection of a monoclonal spike (M spike), often originating from lambda or kappa light variable chains detected by electrophoresis. Interestingly, the kappa variable chain KV37 exhibited the most substantial surge, accompanied by a fold change of 82.6. However, it is essential to note that this elevation of a singular light chain was not universally present among all patients, and did therefore not reach above the statistical threshold.

Differential expression analysis revealed a significant decrease in plasma levels of four components of the complement C1 complex in MM patients (Figure 4A), namely the proteins C1qB, C1qC, C1r and C1s. Interestingly, this observation was specific to MM and was not detected in any other cancer, including the three immune cell malignancies: lymphoma, acute myeloid leukemia (AML), and chronic lymphocytic leukemia (CLL). We could observe this effect in our label free data as well (Appendix A).

The C1 complex, as part of the innate immune system, initializes the classical complement pathway activation. It consists of five components, C1q built up from C1qA (not quantified), C1qB and C1qC [23]. Further components are peptidase C1r and serine protease C1s. Complement activation occurs after the binding of the globular domain of C1q to target molecules, including IgM and IgG (Figure 4B). The binding of C1q initializes the activation of C1r, which in turn leads to C1s activation. The activated C1s initializes the following proteolytic complement cascade, leading to cell lysis, the activation of phagocytes, and the induction of inflammation [23]. The complement system and its activation or suppression have been related to pro- as well as anti-tumoral effects in a wide variety of cancers [24].

### 2.5. Decreased JCHAIN and CD5L Plasma Levels Distinguish Multiple Myeloma

To further identify the unique patterns within the plasma proteome of MM patients we trained a model based on a random forest algorithm to predict disease outcome. We could distinguish MM patients from all other cancer diagnoses with high confidence based on their plasma protein signature (AUC = 0.96) (Figure 5A). Here, the model identified the downregulation of JCHAIN and CD5L plasma as the most powerful proteins to separate MM from 14 other cancer types (Figure 5B). Further proteins that defined the plasma protein signature of MM in our study were the previously described decreased levels of complement proteins C1q, C1r, and C1s as well as upregulated TGFBI, CFD, and MGP and downregulated CBPN (Appendix A). Eight out of the nine of these proteins are linked to the regulation of the complement system and interaction with immunoglobulins. As a joining chain, JCHAIN connects the Fc regions of IgM and IgA and is necessary for the transport of these polymeric immunoglobulins across epithelial cells [25]. JCHAIN-negative IgM has been reported to induce a stronger complement activation than JCHAIN-positive IgM [25,26,27]. 

As quantitative information was not available on either IgM or IgA in the analyzed patients, it was not possible to specify whether decreased JCHAIN levels were accompanied by decreased IgM levels. However, we also found the circulating protein CD5L, also called apoptosis inhibitor of macrophage, to be downregulated. CD5L has been reported be an integral part of IgM and binds to the Fc region of IgM and utilizes immunoglobulin as a carrier which prevents its renal excretion [28,29,30]. We found CD5L and JCHAIN to be highly dependent (Appendix A) as recently shown by Oskam and coworkers [31]. As for TGFBI, CFD, and MGP, MM patients displayed the highest median plasma concentration in comparison to the other cancer patients. TGFBI has been reported to be both tumor-suppressive as well as tumor-promoting in multiple cancers depending on the cancer progression [32].

## 3. Discussion

Cancer is the second-highest cause of mortality in the world. Improved methods that can detect changes in cancer-associated proteins are needed. Our study presents a large pan-cancer initiative in which proteins were absolutely quantified in human plasma using SIS-PrESTs technology. The analytical strategy based on SIS-PrESTs was capable of quantifying proteins with high precision, as they are long polypeptides added as the first step of sample preparation. Therefore, the biological variance across protein targets could be accurately measured. A disadvantage of using internal standards for quantification lies in the fact that they have to be spiked prior to the sample preparation and DIA analysis. Therefore, the availability of targets for absolute quantification can be a limiting factor. However, the DIA strategy allows to acquire all detectable proteins in samples, which can be used to explore the label-free part of the dataset to identify proteins without the precision of absolute quantification. Furthermore, the analytical sensitivity of today’s non-depleted targeted proteomics measurements is limiting. Today’s data acquisition of blood-based tests is comprehensive and has a promising quantitative performance, but the assay is restricted by the dynamic range of plasma, limiting its full potential. Notably, more than 56 FDA- or CLIA-approved biomarkers could be measured in multiplex using this strategy, and by not depleting the plasma, the quantitative integrity of the samples can be assured. This shows that targeted proteomics is an attractive alternative to more sensitive methods based on affinity reagents. 

Within this study, we identified proteins that are implicated in the regulation of complement activation and interaction with immunoglobulins and which we suggest as a plasma biomarker panel for MM. These target proteins include JCHAIN, CD5L and four proteins of the C1 complex. Notably, the protein most predictive for MM in the random forest model was JCHAIN, which links two monomer units of either IgM or IgA together. In the case of IgM, the JCHAIN dimerizes and acts as a nucleating unit for the IgM pentamer. The work of Wang et al. [33] has shown that the CD5L loss turns safe Th17 cells into pathogenic cells, causing autoimmunity. By altering lipids, CD5L affects Rorγt, the master regulator, shifting the immune balance. As CD5L is a major switch of Th17 cell functional states in vivo, this may indicate that Th17 cell functions are dysregulated in Myeloma Patients.

The role of the complement system and its components as potential biomarkers for cancer has been debated in the literature. Here, the role of C1 in cancer has been a double-edged sword. In clear-cell renal-cell carcinoma, the tumor-induced formation of the complement C1 complex in its microenvironment has been described [24]. In contrast, the protein C1q has previously been related to pro-apoptotic and anti-tumor activities in prostate, breast or ovarian cancers [34,35,36]. Furthermore, decreased serum levels of C1q have been described in patients with MM and have been suggested as a potential biomarker for the tumor burden [37]. The systemic decrease of C1q levels in plasma highlighted in our study supports previous findings. Furthermore, we not only observed a decrease of C1q but also of C1r and C1s, suggesting a downregulation of all the proteins forming the C1 complex. Interestingly, this decrease does not extend to other complement proteins, which highlight the proteins related to the C1 complex as possible biomarkers in MM. 

The level of TGFBI has been suggested as a biomarker for tumor progression [32]. Another upregulated protein, CFD, is part of the alternative complement pathway. CFD is a serine protease, which cleaves factor B to form C3-convertase [38]. Whereas decreased CFD levels have been reported in obesity [39], there are scant reports of its direct involvement in cancer. CFD has been suggested as a biomarker for cutaneous squamous-cell carcinoma [40]. Matrix Gla protein (MGP) has been connected to the inhibition of calcification and there is evidence of its relation to the progression of different cancers [41]. Finally, Carboxypeptidase N catalytic chain (CPN1) is part of the Carboxypeptidase N complex, which has been shown to lead to the inactivation of C3a, C4a, and C5a [42,43,44]. CPN has been suggested as a prognostic biomarker in breast cancer and it has been reported that MM patients sensitive to bortezomib treatment have lower CPN levels [45,46]. Therefore, we suggest that these four proteins might not be unique identifiers for the classification of MM patients. However, the overall protein levels of these targets provide an interplay that is unique for MM and requires further investigation. Yet, it must be noted that the majority of these proteins interact with the complement cascade and immunoglobulins.

In conclusion, we describe a targeted proteomics approach capable of measuring hundreds of proteins with their concentrations reported on an absolute scale. This multiplex approach provides a complementary strategy to standardized clinical assays and provides the absolute concentrations of a large number of plasma proteins. Here, we show that this approach can be used with liquid biopsies to identify protein targets which are unique for the detection of multiple myeloma patients.

## 4. Materials and Methods

### 4.1. Ethical Statement

The research adheres to all pertinent ethical guidelines. This pan-cancer study received approval from the Swedish Ethical Review Authority (EPM dnr 2019-00222) and aligned with donor consents in U-CAN (28631533, EPN Uppsala 2010-198 with amendments), with all participants providing written informed consent. The study protocol is in accordance with the ethical principles outlined in the 1975 Declaration of Helsinki.

### 4.2. Cohort

A sample cohort consisting of blood plasma from 1800 cancer patients was provided by the biobank of the Uppsala–Umeå Comprehensive Cancer Consortium (UCAN). The samples were collected following the same protocol. Briefly, blood was collected by venipuncture in 6 mL EDTA tubes (Vacuette Cat. no.456243, Greiner-bio One; Kremsmünster, Austria) and centrifuged at 3000 rcf at room temperature (RT) immediately after sample collection. Plasma was transferred to 0.5 mL tubes and frozen and stored at −80 °C. The plasma samples were fully randomized into thirty-three 96-well plates and deidentified. Plasma from 3 males and 2 females was pooled and added to each plate in triplicate. The cohort consisted of patients diagnosed with one out of fifteen cancers: pituitary neuroendocrine tumors (PIT NET, n = 50), lymphoma (n = 56), chronic lymphocytic leukemia (CLL, n = 50), acute myeloid leukemia (AML, n = 52), multiple myeloma (MM, n = 55), breast cancer (BRC, n = 164), ovarian cancer (OVC, n = 179), endometrial cancer (ENDC, n = 110), cervical cancer (CVX, n = 110), prostate cancer (PRC, n = 172), colorectal cancer (CRC, n = 248), small intestinal neuroendocrine tumor (SI-NET, n = 54), lung cancer (LUNGC, n = 289), meningioma (n = 51), and glioma (n = 160).

### 4.3. Sample Preparation

A set of 276 absolutely quantified SIS-PrESTs of 276 proteins was pooled at close-to-endogenous levels in healthy blood plasma (Appendix A), creating an artificial heavy labeled plasma. The SIS-PrEST pool was aliquoted in 96-well plates and vacuum dried for 16 h at 35 °C and stored at −20 °C. Patient plasma samples were thawed on ice for 1 h and 2 µL diluted 20× with 1× phosphate-buffered saline (PBS, Sigma Aldrich, St. Louis, MI, USA), RapiGest (Waters, Milford, MA, USA), dithiothreitol (DTT, Sigma Aldrich, St. Louis, MI, USA), and diluted plasma corresponding to 1 µL of raw plasma was added to the vacuum-dried SIS-PrESTs to final concentrations of 0.1% RapiGest and 10 mM DTT. The samples were reduced at 37 °C for 1 h and alkylated with 50 mM chloroacetamide (CAA) for 30 min in the dark. Digestion was performed overnight using SOLu-Trypsin (Sigma-Aldrich, St. Louis, MI, USA) in an enzyme:substrate ratio of 1:50 and quenched with trifluoroacetic acid (TFA) to a final concentration of 0.5% (*v*/*v*). Half of each sample was desalted using in-house packed C18 StageTips, according to Rappsilber et al. [47]. In brief, the matrix from 3 layers of Empore C18 disks (Supelco, Sigma Aldrich, St. Louis, MI, USA) was activated with 100% acetonitrile (ACN) and equilibrated with 0.1% TFA. The digest was loaded into the StageTip and washed twice with 80 µL of 0.1% TFA and eluted twice with 30 µL 80% can and 0.1% formic acid (FA). The StageTips were centrifuged for 2 min at 1000 rcf after each addition. Eluted peptides were vacuum dried at 45 °C for 30 min. Prior to analysis, samples were dissolved in Solvent A (3% ACN, 0.1% FA). The samples were processed in batches of two to four plates per digestion.

### 4.4. Mass Spectrometry Analysis

Peptides were quantified in an online system of Ultimate 3000 (Thermo Fisher Scientific, Santa Clara, CA, USA) LC connected to QExactive HF (Thermo Fisher Scientific, Santa Clara, CA, USA) MS. A sample corresponding to 2 ug of raw plasma was loaded onto a trap column (PN 164535, Thermo Fisher Scientific, Santa Clara, CA, USA) and washed for 3 min at 7 µL/min with 100% Solvent A. The peptides were separated on an analytical column (PN ES902, Thermo Fisher Scientific, Santa Clara, CA, USA) using a 40 min linear gradient of 1–32% Solvent B (95% ACN, 0.1% FA) at 0.7 µL/min. The columns were washed with 3 two-minute seesaw gradients of 1–99% Solvent B and equilibrated for 9 min. MS analysis was performed using a DIA method with cycles consisting of a full MS scan (30,000 resolution, AGC = 3 × 10^6^, 300–1200 *m*/*z*, IT = 105 ms) followed by 30 DIA scans (30,000 resolution, AGC = 1 × 10^6^, NCE = 26, 10 *m*/*z* isolation window, IT = 55 ms).

### 4.5. Absolute Quantification

A list of proteotypic peptides was generated by in silico digestion of the fasta file including the amino acid sequences of all 276 spiked-in SIS-PrESTs using EncyclopeDIA (ver. 1.2.2) [48] and whole human proteome as background (Homo Sapiens, UniProt ID: #UP000009606, 20,370 entries, accessed on 26 October 2020). One missed cleavage was allowed, and other parameters were adjusted according to the MS method. A spectral library was generated for all peptides using a Prosit machine learning algorithm [49]. The background proteome and the first 6 analyzed raw files were imported into Skyline (ver. 20.2.0.286) [50] and the peaks were manually inspected. Peptides in which both light and heavy signals were not detected were deleted together with interfering transitions. Peptide retention times were predicted by an indexed retention time library which included the 12 most intensive APOA1 peptides. A so prepared Skyline file was used to import all of the resulting raw files using 3 min windows of predicted peptide retention time with mass accuracy set to 5 ppm. Data for both light and heavy signals were exported for further analysis.

Exported results were imported into RStudio (ver. 1.4.1717). First, the data were filtered to contain only quantified peptides (rdotp > 0.7, dotp > 0.5, 1000 > ratio to standards > 0.01). Samples that the failed APOA1 iRT regression or had fewer than 120 proteins quantified were excluded from the analysis. Additionally, peptides with a quantification rate of less than 50% across all the samples were excluded. Non-paired transitions were filtered out and the heavy to light ratio was calculated from the summed AUCs of transitions present in both light and heavy channels. Further, the data were median-normalized using the pool samples. 

### 4.6. Label-Free Data Extraction

Label-free data was extracted using EncyclopeDIA. First, mzML files were generated from raw files using msConvert within ProteoWizard [51] followed by EncyclopeDIA [52] search against a spectral library generated from list of blood plasma proteins with Prosit integrated into ProteomicsDB [49]. A whole human proteome (Homo Sapiens UniProt ID: # UP000005640, reviewed, 20,371 entries, accessed 11 August 2021) was used as a background proteome.

### 4.7. Disease Prediction

A random forest prediction model was built aiming to classify multiple myeloma patients based on peptide levels in plasma. First, peptide levels for missing values were imputed using the impute.knn function from the impute R package (ver. 1.64.0). The model was built using the train function in the caret R package (ver. 6.0.90) using 70% of the cohort and 5-fold cross validation. The model was tested on the remaining 30% and specificity, sensitivity, and AUC scores were summarized in a receiver operating characteristic (ROC) curve.

## 5. Conclusions

In conclusion, our study pioneers a targeted proteomics approach with a precise quantification of hundreds of plasma proteins, offering a complementary strategy to standard clinical assays. We identified potential biomarkers, including JCHAIN, C1 complex proteins, and others, for multiple myeloma detection. This work underscores the promise of targeted proteomics in cancer diagnostics and biomarker discovery.

## Figures and Tables

**Figure 1 cancers-15-04764-f001:**
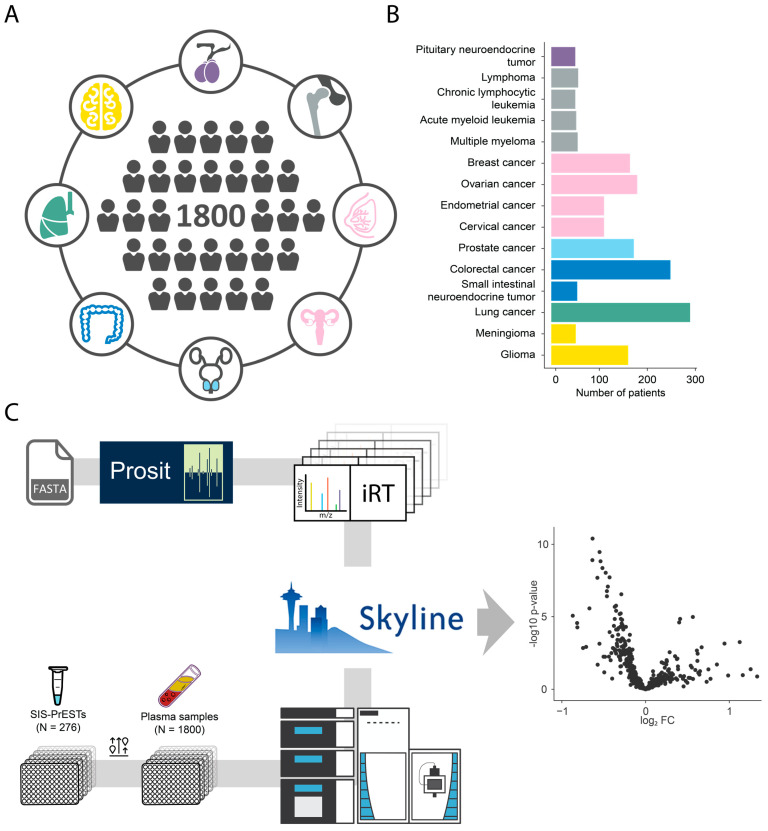
Overview of the pan-cancer cohort and study design. (**A**) Cohort of investigated cancer patients (n = 1800) distributed across eight different organs (each color highlights tissue origin of the cancer). (**B**) Distribution of patient samples (n = 1800) across fifteen different cancer subtypes. (**C**) Overview of analytical workflow, which includes library generation (top row) and flow of patient samples with spike in SIS-PrESTs (bottom row) prior to chromatography extraction and data analysis (middle).

**Figure 2 cancers-15-04764-f002:**
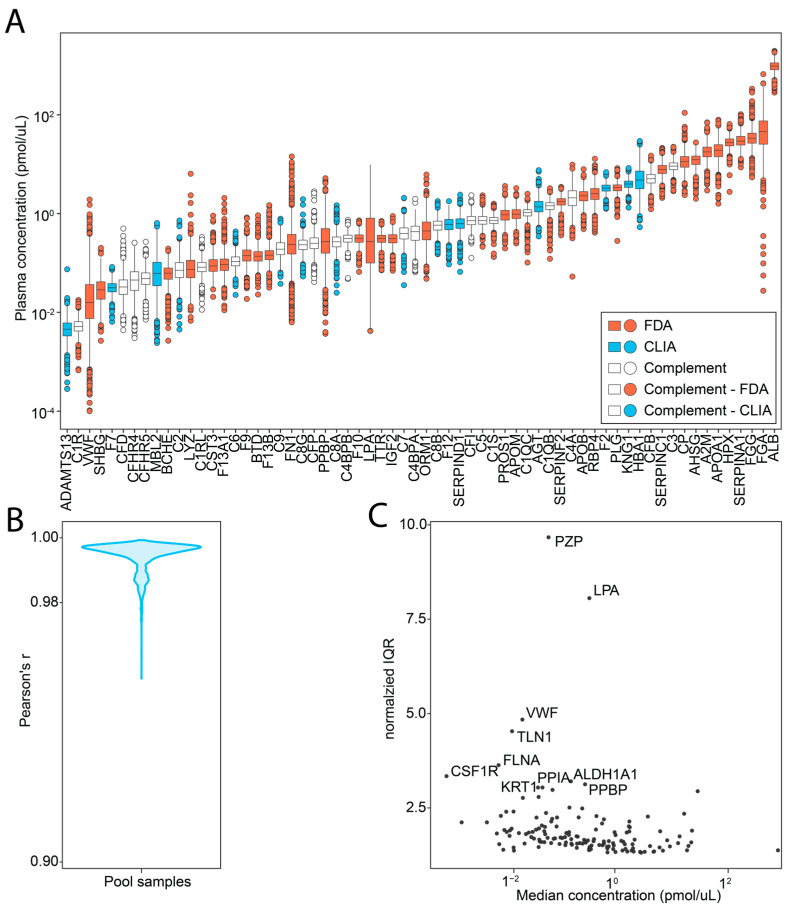
Proteins quantified in the targeted proteomics workflow. (**A**) The dynamic range and concentration (y-axis) of all FDA (red) and CLIA (blue) markers, supplemented with complement system proteins (white) (x-axis). (**B**) Density distribution of the cross-correlation (Pearson’s r) between all pooled technical replicates in the dataset. (**C**) Inter-individual variation in the human plasma proteome visualized as the normalized interquartile range (IQR, y-axis) plotted versus the median protein concentration (pmol/µL, x-axis).

**Figure 3 cancers-15-04764-f003:**
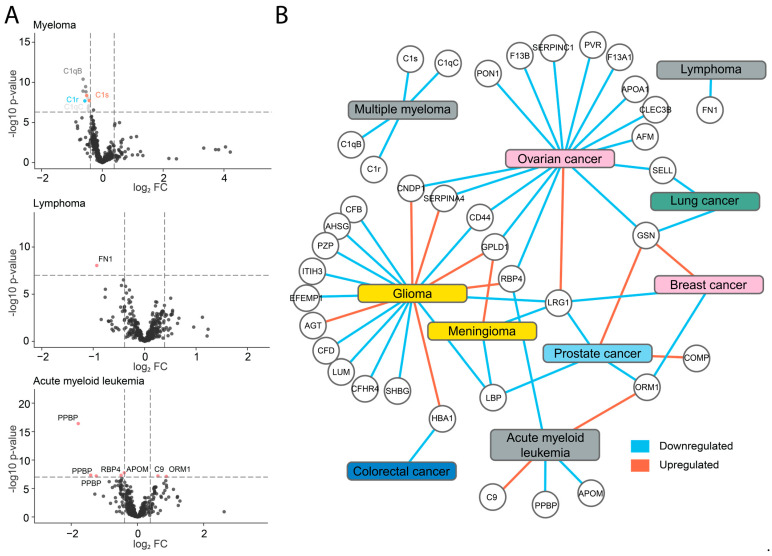
Identification of differentially expressed proteins in the pan-cancer cohort. (**A**) Comparisons of plasma protein levels between myeloma, lymphoma, and acute myeloid leukemia versus all other cancers group (Student’s *t*-test), visualized in volcano plots with labels on the topmost significant proteins. (**B**) Network visualization of all protein targets identified as differentially expressed in one cancer compared to all the other cancers (*p*-value < 0.0005, Bonferroni adj.). Blue and red connections signify up- or downregulation, respectively. Each cancer is colored by tissue of origin.

**Figure 4 cancers-15-04764-f004:**
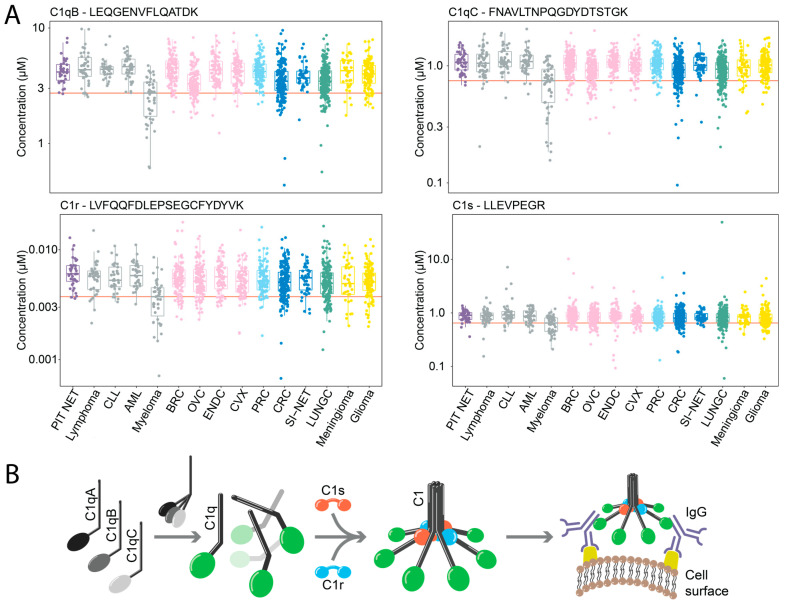
Quantification of the complement system and its circulating components across 15 different cancers. (**A**) Boxplots that visualize the protein levels (y-axis, log2 concentration) of C1qB, C1qC, C1r and C1s across all patients (n = 1800), grouped by cancer type (y-axis) and colored by tissue origin. (**B**) General overview of the C1 complex with genes involved in its architecture.

**Figure 5 cancers-15-04764-f005:**
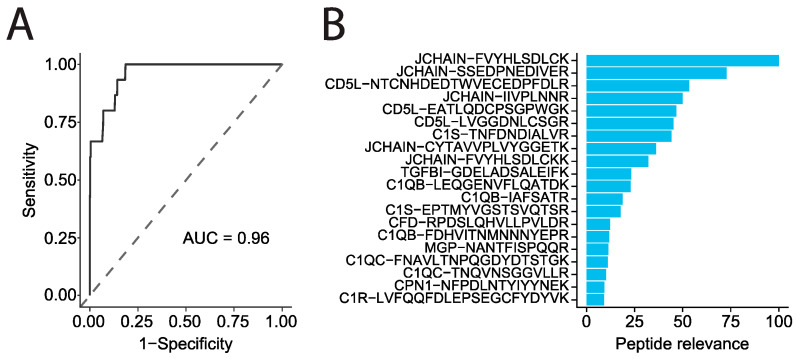
Data-driven analysis reveals a panel of biomarkers suitable for MM prediction. (**A**) ROC curve visualizing the performance and classification specificity of a model generated by a random forest algorithm, which was used to distinguish MM from all other fourteen cancers, with an AUC of 0.96. (**B**) Feature selection based on peptides, visualizing their relevance score (y-axis) and impact on the model used to separate MM from the fourteen other cancers (color intensity represents the impact level of each analyte, ranging from 0 to 100).

## Data Availability

The MS raw data as well as Skyline files and libraries are available on Panorama public “https://panoramaweb.org/QKcywA.url (accessed on 19 September 2023) and ProteomeXchange, ID PXD037946.

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
