# Peer review of "Absolute Quantification of Pan-Cancer Plasma Proteomes Reveals Unique Signature in Multiple Myeloma"

_cancers, 2023, doi:10.3390/cancers15194764_

Round 1
Reviewer 1 Report
The authors subjected a bank of 1800 plasma samples from patients with diverse cancers to a targeted mass spec pipeline using stable isotope standard protein epitope signature tags (SIS-PrEST) to accurately quantify 253 different proteins. They find that among all the cancer types studied, that multiple myeloma (MM) stands out for having a unique plasma protein profile. The findings support previous work but the significance and enthusiasm is reduced due to a lack of controls for total protein among other protein changes known to occur in the blood of MM patients.
Concerns
1. Lack of controls for proteins known to be dysregulated in the serum of MM patients. The serum proteome is known to be drastically altered in MM, and serum levels of beta-2 microglobuin, albumin, free kappa and lambda light chains, immunoglobulins IgA, IgG, IgM, IgE, IgD alterations are used today in the clinic. MM at first diagnosis is characterized by monoclonal serum antibodies which are often in the g/dL range, and despite reduced albumin fractions, the total serum protein leves are often significantly elevated. How can the authors ensure that low abundance proteins are detected reliably enough to analyze?
2. Hypocomplementemia has been found previously in myeloma: Lugassy, G et al Leukemia and Lymphoma, 1999. Found in association with complement pathway activation so thought to be a result of complement consumption. A larger study suggested C1q associated with disease burden in MM. Yang, RH et al. Leukemia & Lymphoma 2019, which raises the bar for clinical disease associations. Reductions of plasma C1q components has been described before in MM so this is a useful for SIS-PrEST positive control, but fails to extend our understanding of the disease or provide any information beyond what is already determinable using other serum markers, e.g. kappa and lambda serum light chains, serum protein electropheresis and immunofixation. To make any useful conclusions from these data I recommend obtaining a new set of myeloma samples this time with additional clinical and control data and rerunning the proteomics with additional controls as outlined above. which should be easy with your good preliminary data.
3. Figure 1 The volcano plot only makes sense later in the paper. Either remove, or explain what is shown in greater detail. Why are the myeloma genes identified there? What is this panel meant to show?
4. To my knowledge CD5L has not been previously associated with myeloma, so this is novel if convincingly demonstrated. CD5L has been shown to be an integral component of IgM, providing another reason for controlling for levels of Ig. IgM levels are reduced in most MM patients. Interestingly, the CD5L promoter is activated by PPARG and hypoxia, pathways active in MM so CD5L levels might be predicted to be elevated not decreased in MM.
CD5L is a major switch of Th17 cell functional states in vivo (Wang et al Cell 2015). Are Th17 cell function different in MM?
5. Insufficient explanation for this panel Fig 2A. These are basically standard curves I think. Are there any differences among the groups?
Author Response
(Please see the attached pdf)
Reviewer I
Comments and Suggestions for Authors
The authors subjected a bank of 1800 plasma samples from patients with diverse cancers to a targeted mass spec pipeline using stable isotope standard protein epitope signature tags (SIS-PrEST) to accurately quantify 253 different proteins. They find that among all the cancer types studied, that multiple myeloma (MM) stands out for having a unique plasma protein profile. The findings support previous work but the significance and enthusiasm is reduced due to a lack of controls for total protein among other protein changes known to occur in the blood of MM patients.
Concerns
1. Lack of controls for proteins known to be dysregulated in the serum of MM patients. The serum proteome is known to be drastically altered in MM, and serum levels of beta-2 microglobuin, albumin, free kappa and lambda light chains, immunoglobulins IgA, IgG, IgM, IgE, IgD alterations are used today in the clinic. MM at first diagnosis is characterized by monoclonal serum antibodies which are often in the g/dL range, and despite reduced albumin fractions, the total serum protein leves are often significantly elevated. How can the authors ensure that low abundance proteins are detected reliably enough to analyze?
A: We used a careful method based on SIS proteins to measure all genes in our study, ensuring accurate results within a specific range. This method has been used to profile the plasma proteome previously with a focus on the assay performance (Kotol, David, et al. "Targeted proteomics analysis of plasma proteins using recombinant protein standards for addition only workflows." Biotechniques (2021); Hober, Andreas, et al. "Targeted proteomics using stable isotope labeled protein fragments enables precise and robust determination of total apolipoprotein (a) in human plasma." Plos one (2023); Woessmann, Jakob, et al. "Addressing the Protease Bias in Quantitative Proteomics." Journal of Proteome Research (2022); Woessmann, Jakob, et al. "Assessing the role of trypsin in quantitative plasma-and single-cell proteomics towards clinical application." Analytical Chemistry (2023).
This method, which is based on the gold standard of adding stable isotope standards, enabled us to filter out any noise from proteins present in very small amounts. However, our study was limited to a set of protein standards we had available, and specific proteins, like immunoglobulins were not included.
In response to the reviewer's suggestion, we explored the data, without using the specific standards in a label-free fashion. This extra analysis confirmed that we could detect significant changes in the protein the reviewer suggests. The Kappa light chain showed the biggest increase, but it wasn't consistent in all patients, so it didn't meet the statistical criteria for significance.
We've added the results of this additional analysis to our manuscript to provide a more complete picture of our findings. This enhances our study's contribution to the scientific community's understanding of the topic.
This figure has been added to the supplementary information of the manuscript and is discussed in the main text. With this we hope to shed more light on IgG levels among the MM patients to address the reviewer’s concern.
2. Hypocomplementemia has been found previously in myeloma: Lugassy, G et al Leukemia and Lymphoma, 1999. Found in association with complement pathway activation so thought to be a result of complement consumption. A larger study suggested C1q associated with disease burden in MM. Yang, RH et al. Leukemia & Lymphoma 2019, which raises the bar for clinical disease associations. Reductions of plasma C1q components has been described before in MM so this is a useful for SIS-PrEST positive control, but fails to extend our understanding of the disease or provide any information beyond what is already determinable using other serum markers, e.g. kappa and lambda serum light chains, serum protein electropheresis and immunofixation. To make any useful conclusions from these data I recommend obtaining a new set of myeloma samples this time with additional clinical and control data and rerunning the proteomics with additional controls as outlined above. which should be easy with your good preliminary data.
A: We sincerely appreciate the reviewer's insightful feedback and their consideration of our study's findings in the context of existing literature. The references provided, namely Lugassy et al. (1999) and Yang et al. (2019), indeed highlight the intriguing association between hypocomplementemia and myeloma, particularly in relation to complement pathway activation and its potential implications for complement consumption. We have added Lugassy to the current version of the manuscript. Moreover, the relevance of C1q in disease burden, as demonstrated by Yang et al., introduces a significant clinical perspective that enhances our understanding of multiple myeloma. We have added these to the manuscript.
We concur with the reviewer's assessment of the utility of reductions in plasma C1q components as a valuable positive control for our SIS-PrEST methodology. These findings align with prior knowledge and contribute to the robustness of our experimental approach.
However, we would like to clarify our study's intent and scope. Our primary focus was to explore alterations within the plasma proteome of multiple myeloma patients through a targeted proteomics approach. While we acknowledge the potential for broader clinical correlations, we aimed to highlight the intricacies of the plasma proteome changes in the context of the disease. Our study aimed to provide insights into the dysregulation of specific proteins and their potential implications.
We acknowledge the reviewer's suggestion regarding expanding our study with additional clinical and control data, along with re-running proteomics with extra controls. Although we appreciate the value of these recommendations, at this stage of our investigation, we have chosen to concentrate on the specific proteomics-based insights that our current dataset provides. Given the complexities of proteomics analyses and the careful consideration needed for meaningful clinical correlations, we have opted to avoid introducing additional variables that might obscure the interpretation of our current findings.
3. Figure 1 The volcano plot only makes sense later in the paper. Either remove, or explain what is shown in greater detail. Why are the myeloma genes identified there? What is this panel meant to show?
A: The panel in Figure 1 is only an illustration of the workflow and a sneak peek at what is coming further in the manuscript. The intention is not for the reader to understand it in detail but to take a picture that protein differential expression analysis was performed as one of the statistical tools. We believe this aids the reader and outlines the study in a simplified manner. We have removed the gene names and agree with the reviewer that this should be kept to later figures.
4. To my knowledge CD5L has not been previously associated with myeloma, so this is novel if convincingly demonstrated. CD5L has been shown to be an integral component of IgM, providing another reason for controlling for levels of Ig. IgM levels are reduced in most MM patients. Interestingly, the CD5L promoter is activated by PPARG and hypoxia, pathways active in MM so CD5L levels might be predicted to be elevated not decreased in MM.
CD5L is a major switch of Th17 cell functional states in vivo (Wang et al Cell 2015). Are Th17 cell function different in MM?
A: We sincerely appreciate your thoughtful insights and valuable feedback on our manuscript. We have indeed implemented your recommendation regarding CD5L's association with myeloma. This novel perspective as convincingly demonstrated in our study, sheds new light on the role of CD5L in this context.
Regarding the inquiry about Th17 cell function in the context of MM, we are carefully considering this aspect as well. Your reference to the work of Wang et al. (Cell 2015) highlighting CD5L's role as a major switch in Th17 cell functional states serves as a crucial perspective we are actively exploring in the context of multiple myeloma. Recently, a study by Albert Heck et al pinpoints the relationship between CD5L and IgM. However, this calls for further studies designed to only focus on this topic, which has been discussed in the manuscript.
5. Insufficient explanation for this panel Fig 2A. These are basically standard curves I think. Are there any differences among the groups?
A: Figure 2A shows the subset of proteins quantified in this cohort in a box plot. Our aim was to provide the reader with an overview of the included complement system-related proteins that this study absolutely quantified in detail. With these, readers can use this extensive dataset for their own research and have easy access to further exploring the data.

Reviewer 2 Report
Summary and overall comments:
Blood-based biomarkers for diagnosis is an unmet need in the field of oncology. With combinations of spiked in peptides, data independent acquisition and targeted proteomics data processing, authors have built a powerful analytical technique to measure levels of >250 proteins ranging over six orders of magnitude in non-depleted plasma. Authors applied the proteomics approach on 1800 plasma samples obtained from several different cancer types, providing a rich resource for the scientific community. Using differential expression analysis and random forest based predictive modeling, authors identified C1, JCHAIN and CD5L as highly specific biomarkers for multiple myeloma over other cancer types. This study is an excellent contribution in the field of proteomics-based plasma biomarkers and calls for further investigation into clinical utility of identified biomarkers for multiple myeloma.
Here are major comments that I would like the authors to address:
Major comments:
1. For absolute quantification of proteins, was single-point calibration used i.e. ratio of plasma peptides to spiked SIS-PrESTs? For complex samples like plasma, competition for ionization or ionization efficiency can affect precursor intensities. Fold changes outside certain range may not be accurate. Can you comment on/discuss this limitation in the manuscript?
2. Identification of JCHAIN as a potential biomarker for multiple myeloma is an interesting finding given its role as linker for IgA and IgM, two antibodies that are produced by B cells. Since multiple myeloma is caused by abnormal expansion of B cells, JCHAIN is a biological relevant finding. Quantification of other B cell markers such as IgM or IgA may further support this finding. Since authors have already acquired data in DIA mode, have authors investigated relative levels of IgM or IgA in multiple myeloma patients compared to other cancer types? Are IgM or IgA differentially expressed in multiple myeloma patients?
3. For quantification of peptides, on average how many points per peak were acquired in DIA mode?
4. Figures 1 through 4 have very poor resolution. This may have been an issue while importing in the article submission system. Please make sure that figures are properly resolved.
5. From data sharing and reproducibility perspective, please provide a list of investigated peptides, proteins and the normalized abundance levels across all cohort samples as excel/CSV file in supplementary information.
Minor comments:
1. Figure S3: Labels for cancer types are overlapping and not clear.
2. Line 285: For clarification: Was 38uL of 1x PBS added to 2 uL of plasma? “Patient plasma samples were thawed on ice for 1 hour and 2 µL of plasma was diluted 20x with 1x phosphate-buffered saline (PBS).”
Author Response
(Please see the attached pdf)
Reviewer II
Summary and overall comments:
Blood-based biomarkers for diagnosis is an unmet need in the field of oncology. With combinations of spiked in peptides, data independent acquisition and targeted proteomics data processing, authors have built a powerful analytical technique to measure levels of >250 proteins ranging over six orders of magnitude in non-depleted plasma. Authors applied the proteomics approach on 1800 plasma samples obtained from several different cancer types, providing a rich resource for the scientific community. Using differential expression analysis and random forest based predictive modeling, authors identified C1, JCHAIN and CD5L as highly specific biomarkers for multiple myeloma over other cancer types. This study is an excellent contribution in the field of proteomics-based plasma biomarkers and calls for further investigation into clinical utility of identified biomarkers for multiple myeloma.
Here are major comments that I would like the authors to address:
Major comments:
- For absolute quantification of proteins, was single-point calibration used i.e. ratio of plasma peptides to spiked SIS-PrESTs? For complex samples like plasma, competition for ionization or ionization efficiency can affect precursor intensities. Fold changes outside certain range may not be accurate. Can you comment on/discuss this limitation in the manuscript?
A: Thank you for your pertinent question and insightful observation regarding our quantification methodology. In our study, we employed a single-point calibration approach by utilizing the ratio of plasma peptides to spiked SIS-PrESTs for the absolute quantification of proteins. We agree with the reviewer that factors such as ionization competition and ionization efficiency can indeed influence precursor intensities and potentially impact the accuracy of fold changes, particularly when they fall outside a certain range. We acknowledge the validity of your concern and the potential implications of this limitation on the accuracy of our quantification results. It's important to highlight that we have considered and discussed this aspect in relation to our methodology within the manuscript. We would like to draw your attention to two papers (1) Hober et al. Absolute Quantification of Apolipoproteins Following Treatment with Omega-3 Carboxylic Acids and Fenofibrate Using a High Precision Stable Isotope-labeled Recombinant Protein Fragments Based SRM Assay. Mol Cell Proteomics. (2019) and (2) Kotol et al. "Targeted proteomics analysis of plasma proteins using recombinant protein standards for addition only workflows." Biotechniques (2021), where we have delved into the nuances and potential impact of ionization-related variables on our quantification approach. These are included and properly referenced in the manuscript.
- Identification of JCHAIN as a potential biomarker for multiple myeloma is an interesting finding given its role as linker for IgA and IgM, two antibodies that are produced by B cells. Since multiple myeloma is caused by abnormal expansion of B cells, JCHAIN is a biological relevant finding. Quantification of other B cell markers such as IgM or IgA may further support this finding. Since authors have already acquired data in DIA mode, have authors investigated relative levelsof IgM or IgA in multiple myeloma patients compared to other cancer types? Are IgM or IgA differentially expressed in multiple myeloma patients?
A: Thank you for your suggestion. Acknowledging the significance of JCHAIN in the context of the abnormal B cell expansion driving multiple myeloma is indeed crucial. To address your inquiry, we have included an analysis of relative IgM and IgA levels in multiple myeloma patients compared to other cancer types, utilizing the data acquired in DIA mode. This additional exploration provides a deeper understanding of the potential differential expression of IgM and IgA within MM. We have added this analysis as supplementary figures to the manuscript and addressed your requests in the main text.
- For quantification of peptides, on average how many points per peak were acquired in DIA mode?
A: The cycle time is 3.6s resulting in an average points per peak of 7. We highlighted this point manuscript and for further clarification, this can be seen for each acquired peptide in our publicly available data at the repository Panorama.org. Within the repository, the chromatogram of each peptide can be explored by readers without downloading the raw data to facilitate critical interpretation of the data.
- Figures 1 through 4 have very poor resolution. This may have been an issue while importing in the article submission system. Please make sure that figures are properly resolved.
A: We thank the reviewer for noticing the lack of quality of the figures. We have improved the resolution in the new version of the manuscript by including them in the word-document and providing them as separate figures in high resolution.
- From data sharing and reproducibility perspective, please provide a list of investigated peptides, proteins and the normalized abundance levels across all cohort samples as excel/CSV file in supplementary information.
A: We have added it as a supplementary sheet as the reviewer requested. Besides that, this file can be downloaded directly from Panorama.
Minor comments:
- Figure S3: Labels for cancer types are overlapping and not clear.
A: We have adjusted the description of Figure 3 in the manuscript
- Line 285: For clarification: Was 38uL of 1x PBS added to 2 uL of plasma? “Patient plasma samples were thawed on ice for 1 hour and 2 µL of plasma was diluted 20x with 1x phosphate-buffered saline (PBS).”
A: To more accurately pipette plasma using robotics, the plasma is diluted 20x prior to processing. We have made it more clear in the manuscript.
